# Human capital, gender, institutional environment and research funding: Determinants of research productivity in German psychology

**Martin Schröder** [1]*, **Isabel M. Habicht**[2], **Mark Lutter**[2]

**1** Faculty of Human and Business Sciences, Saarland University, Saarbrücken, Germany, **2** Institute of Sociology, University of Wuppertal, Wuppertal, Germany

* martin.schroeder@uni-saarland.de

**Data Availability Statement:** We have provided all data publicly, together with the files to replicate all our analyses under the URL: https://osf.io/qdnb2/?

## Abstract

Which academics are more productive? The "sacred spark" theory predicts that some researchers are innately more productive than others, while the theory of cumulative advantage argues that small initial inequalities accumulate to large differences in productivity over time. Using a virtually complete panel dataset of all academic psychologists found in German universities in 2019, including their career information and publications, we examine under what conditions male and female psychologists publish more peer-reviewed articles. The strongest predictor of this is prior experience in publishing peer reviewed journal articles, irrespective of other prior endowments. This relationship between earlier and later productivity is not strongly confounded by career stage, affiliation with elite institutions, receipt of third-party funding, or parenthood. The effect of prior publications on current productivity explains why female academic psychologists publish less than men do. While female psychologists publish 34% less than their male counterparts, this gap diminishes to 17% after controlling for prior publication experience. This lends supports to the theory of cumulative advantage, which explains overall differences in productivity over entire careers by the accumulation of minor initial inequalities to large outcome differences over time.

## Introduction

Few scholars would disagree with Robert Merton's [1] claim that "[t]he institutional goal of science is the extension of certified knowledge." Many would concur that the extension of certified knowledge is best achieved through publication of peer-reviewed journal articles, which can therefore be used to gauge a scientist's productivity [2–6]. But why are some scientists more productive in publishing such articles? Do they possess something akin to a "sacred spark", which makes them inherently more productive, irrespective of prior training [7]? Or do researchers start at relatively similar productivity levels, differentiated only by minor initial factors, which then, however, accumulate to large outcome differences over whole careers [for a review, see 8]? This study uses a virtually complete sample of all academic psychologists at

view_only=dd06a0a9adf54f8e8d85992bc660bdda
The only exception is information on the number of
children, as this is private information, which—
under German law—cannot be provided publicly.

**Funding:** Martin Schröder: BMBF (German Federal
Ministry of Education and Research) third-party
top-up funding for proposal: "Constraints for
scientists due to the Corona pandemic and impact
on publication performance" (funding code
01PU17015A). Martin Schröder: BMBF (German
Federal Ministry of Education and Research) third-
party funding for proposal: „Measuring scientific
productivity, its preconditions and consequences"
(funding code 01PU17015A). Mark Lutter: BMBF
(German Federal Ministry of Education and
Research) third-party funding for proposal:
„Measuring scientific productivity, its preconditions
and consequences" (funding code 01PU17015B).
Funder website: https://www.bmbf.de/bmbf/en/
home/home_node.html The funder played no role
in the study design, data collection and analysis,
decision to publish, or preparation of the
manuscript.

**Competing interests:** The authors have declared
that no competing interests exist.

German universities to examine who is particularly productive in publishing peer-reviewed journal articles.

Understanding productivity in academic psychology is crucial for a number of reasons. First, psychology has a fairly equal initial gender representation overall. Yet while its academic career pipeline starts with a high proportion of female graduates, it ends with more male professors [9, 10]. Psychology is thus not a field that women shun generally, yet few become tenured professors. This poses the question why women fall behind men over successive career stages. Secondly, while psychology is similar to other social sciences in this regard, its performance standards are oriented towards the natural sciences, as research output is chiefly measured through publications in internationally visible peer reviewed journals. Understanding who is more productive in publishing such articles may therefore shed light on differences in a field with similar initial numbers of men and women. This initial gender parity is an advantage over many natural sciences; and the measurability of productivity based on a widely accepted metric is an advantage over many social sciences.

Our results suggest that psychologists become much more productive with prior publication experience. This is one reason why women increasingly fall behind, as they have fewer initial publications to build on, and these minor initial differences accumulate to large productivity differences over time. In the following, we discuss the theoretical mechanisms and hypotheses behind these results.

## Theory: What explains productivity?

### Human capital accumulation

Human capital is the sum of knowledge, experience and skills accumulated throughout a career [11]. In academia, human capital should be related to career progression, which is measured through degrees such as a PhD or a tenured professorship. Yet apart from such signs of certified knowledge, on-the-job training also exists, as publishing brings experience that presumably helps the early career academic to publish more in the future.

This entails an endogeneity problem, as research productivity is not merely an output of past research, but also an input for future productivity. That past success lays the foundation for future success is known as the "Matthew Effect" [12], which the theory of "cumulative advantage" has conceptualized [7, for a review of the literature, see 8]. The opposite of this is the "sacred spark" hypothesis, which argues for innate differences between academics in talent, drive and creativity. This latter hypothesis contradicts the theory of cumulative advantage by suggesting that differences in productivity exist regardless of prior experience [7]. While the theory of cumulative advantage thus suggests that research output can be explained by a researcher's accumulated publication experience, the sacred spark hypothesis instead suggests that some researchers have higher rates of research output, even with the same prior experience.

### Confounder: Gender

Crucially, women are said to accumulate human capital in academia more gradually than men. The starting point for this observation is that women are generally found to produce fewer publications than their male counterparts [2, 5, 13, 14], which is also the case in psychology [5, 10, 15]. This may be due to a gendered cumulative advantage in psychology [9]. Notably, scholars [16] argue that "women scientists publish fewer papers than men because women are less likely than men to have the personal characteristics, structural positions, and facilitating resources that are conducive to publication." This suggests that women suffer from a cumulative disadvantage. For example, women may start their careers with PhDs from less prestigious

institutions [3], which may have little impact on their early productivity. Yet their lower-prestige PhDs may subsequently hinder women from accessing higher quality postdoc institutions, which then hinders later publications, so that small initial disadvantages accumulate to large productivity differences over whole careers [17]. Another prominent explanation for why women accumulate success more slowly is childrearing [4, 10, 18, 19], as parental obligations may decrease the productivity of women more than of men [though cf. critically, 5; for an overview, see 18].

Early research has suggested that cumulative falling behind due to childrearing and other gendered factors may explain 60% of the gender productivity gap, while 40% of the differences in career progression cannot be explained using prior resource endowments [20]. However, the cross-sectional data used so far by research in this area cannot explain whether differences persist irrespective of prior experience or whether lower initial endowments accumulate to large differences over time [5]. Another problem is that some research does *not* argue that academics become more productive with successive career steps, but rather that productivity eventually declines, even suggesting the opposite of a process of cumulative advantage [2, 5, 21].

While some, therefore, argue that prior productivity predicts current productivity [22], it is unclear not only whether this is actually the case, but also whether the female productivity gap can be ascribed to cumulatively falling behind, as the theory of cumulative advantage suggests. Another problem is that the few existing studies come from STEM disciplines, leaving unclear whether a mechanism of cumulative advantage explains productivity of academic psychologists [23]. We will, therefore, examine the sacred spark versus cumulative advantage hypothesis by analyzing first to what degree earlier publications and endowments explain later productivity (which would confirm the cumulative advantage hypothesis), and then to what degree current productivity exists irrespective of past endowments (which would confirm the sacred spark hypothesis). To understand whether women fall behind cumulatively, we test whether women publish less than men before and after holding prior experience constant. However, this must take account of further confounding variables that can be seen as affecting research productivity, notably a researcher's institutional environment and research funding.

## Confounder: Institutional environment

High-status universities grant psychologists better access to resources, such as training, peers and mentors. This makes institutional environment an important confounder in how well a researcher can translate her human capital into research output. Germany's so-called "Excellence Initiative" has bestowed the title "university of excellence" on some universities that claim to support researchers for maximum productivity [24]. Whether this is true remains unclear, however, as no studies show whether "universities of excellence" actually foster more productive researchers [25]. If they do, then "universities of excellence" would be an important mechanism behind cumulative advantage, making those researchers more productive who—presumably—were in the first place chosen for their higher productivity. It is also possible however, that while more productive researcher may be more likely to gain access to these institutions, being at these institutions might not in itself increase their publication rate, which would suggest that no cumulative advantage arises from membership of these institutions as such.

Yet while some studies suggest that more prestigious departments do indeed make scholars more productive [26], others are skeptical [2]. While some [27] argue for a strong link between an institution's publication output and research expenditure, this leaves the question unanswered whether the same individual becomes more productive after visiting a better-endowed

institution or whether these institutions attract those who are more productive in the first place—i.e. rather than making them more productive [28, 29].

A similar problem arises when trying to measure the effect of international mobility on research productivity. International mobility may enhance one's career prospects due to network expansion or exposition to new and stimulating environments. Yet international mobility may itself be endogenous to research productivity, as more productive academics may have better access to high-status institutions abroad [30]. This is one reason why the effect of international experience on productivity is unclear: some studies reject such an effect [31], while others suggest one [30]. To disentangle the effect of international experience on productivity, one needs to first measure the relationship between research output and prior research experience, and then test whether this relationship holds after controlling for experience in different institutional environments, disentangling whether international experience acts as a confounder that increases a researcher's output beyond what would be expected given her prior publication trajectory.

### Confounder: Research funding

Another confounder on the link between earlier and later research productivity is research funding, which can be a resource to increase research output beyond what would be expected based on a researcher's prior publication trajectory. Yet whether research funding actually increases research output is debated [5], with some suggesting substantial effects [32], and others rejecting this [28]. Therefore, even studies that do find effects ask for a replication of their findings in different scientific disciplines with longitudinal data [32]. This again needs to address reverse causality, as funding may not only result in, but also result from prior publications [33]. It is therefore important to first test whether some psychologists acquire more funding in the first place, and then test whether such funding explains current research productivity beyond what would be expected given prior research productivity [34].

### Data and methods

In 2019, six supervised research assistants worked 19 hours weekly for one year to code all CV and publication data from the websites of the two relevant Max Planck Institutes in psychology, as well as those of all 72 German universities with a psychology department. Since universities of applied sciences are mainly oriented towards teaching rather than research, our selection provides a virtually complete career dataset of Germany's 2,528 research-active academic psychologists who received their PhD after 1980. Each publication trajectory starts with the year of an individual's first publication and ends with the last publication found at the time of coding, leading to retrospective publication trajectories with 25,828 researcher-years. We lag all predictors by one year, as effects on productivity need time to unfold, and also to avoid simultaneity bias. This reduces the usable dataset to 23,300 researcher-years, containing 1,191 female and 984 male psychologists and 10,528 female and 12,772 male research-years. We added data about each individual's research funding from the website of the German Research Foundation (DFG). We also conducted an email survey to assess parental status. Of all male psychologists, 56% answered, while 65% all female psychologists responded. All information was anonymized, so individual researchers cannot be tracked. Our data and do files to replicate our analyses are available online (https://osf.io/qdnb2/?view_only= dd06a0a9adf54f8e8d85992bc660bdda).

### Variables

Tables A1–A4 in the S1 Appendix show descriptive information on all used variables, separately for men and women at each of four possible career stages (pre-PhD, postdoc, untenured

professor, tenured professor). Our dependent variable is a psychologist's count of annual publications in peer-reviewed journals listed in the Social Science (SSCI) or Science Citation Index Expanded (SCIE). Studies of psychology have shown that such journal articles are the most relevant measure of productivity in academic psychology, leading to tenured professorships like no other measure of scientific productivity [35]; SSCI/SCIE articles are also highly correlated to other types of productivity, further making it reasonable to assume that they are a good indicator of general research productivity [2, 36]. We adjust journal articles for co-authorship, using the weighting factor *2/(# of authors+1)*. This way, a co-authored publication counts as 0.67 of an individual publication, a publication with three authors counts as 0.5, etc. We also calculated co-author adjusted publications by the formula (1/# of authors) as done by others [37, 38]; but our results are robust to this (see Table A5 Model 2, compared to Table A5 Model 1 in the S1 Appendix) and even to counting each publication as single-authored regardless of the number of co-authors (see Table A5 column 3 compared to column 1 in the S1 Appendix). We also weighted each SSCI/SCIE article by the journal's impact factor, which did not substantially change our results either (cf. Table A5 Model 4 compared to Model 1 in the S1 Appendix), suggesting that our dependent variable of publishing peer reviewed articles is robust to the prestige of different journals.

Not all academics show their entire publication list. Senior academics in particular tend to report their top publications only. We account for this missing data by tagging researchers with "selected publication" lists through a corresponding dummy. Incomplete publications lists are fairly equally distributed across gender (around 5% of women and 9% of men, see Table A7 in the S1 Appendix), so we do not assume that our results are biased by gender-specific reporting. To adjust our models for varying labor market conditions, we use dummy variables for cohorts, based on when individuals had their first publication. To adjust for prior productivity, we control for six types of accumulated prior research output (each variable is adjusted for co-authors as described above): the number of 1) prior SSCI/SCIE journal articles, 2) monographs, 3) book chapters. 4) non-SSCI/SCIE journal articles, 5) edited volumes, 6) other literature (gray literature such as reports, working papers, literature reviews). While other researchers [39] have used a combined productivity metric, we adjust for each type of prior productivity separately. We add the constant of 1 and then log all publications as well as other continuous variables to account for diminishing returns, since e.g. publishing a sixth article increases research output by 20% compared to the fifth, while publishing a second article increases research output by 100% compared to the first article.

A female dummy accounts for gender. Dummy variables also account for the following career stages: 1) pre-doc (no PhD yet), 2) post-doc (PhD, but no *Habilitation—which is a post-doctoral thesis in the German academic system—*or assistant professorship), 3) assistant professor (*Habilitation* or assistant professorship but no tenured professorship), 4) tenured professorship (W2/W3, which are rofessorial salary grades in German academic system). Further splitting up the category of tenured professorship into the categories W2 and W3 made little sense, as differences between types of tenured professorships are typically minor. We separately measure career steps at a "university of excellence"; we measure international experience through a doctoral degree from an institution obtained outside Germany and through the number of months spent abroad, assuming five months for a semester and ten months for an academic year if websites did not list exact months. Research funding is measured through a researcher's grants from Germany's largest funding agency DFG (accessed through https://gepris.dfg.de/).

Data from our email survey allows us to measure whether researchers have children. We separate this information by gender, to account for motherhood and fatherhood separately. These variables thus have time-varying coefficients that take the value of 1 when a

psychologist's first child is born. The status "children unknown" accounts for non-response by gender. We conducted a complete-record analysis, using only information from those survey-participants who responded to our question about children. The results for this group are virtually the same as in our main specification, suggesting that lack of information on children does not systematically bias our results (see Table A5 Model 5 compared to Model 1 in the S1 Appendix).

All analyses were done using Stata. The following sections first describe the correlates of research output and then present the results of random-effect (RE) models, which—as a first step—explicitly do *not* control for prior publications. These regressions are therefore still descriptive in the sense that they show how career trajectories differ among psychologists. We then use RE regressions that *do* control for prior research output, thus showing who is more productive than others, irrespective of past productivity. As these models adjust for prior productivity, while the previous models did not, the difference between these and the prior RE models shows whether the productivity of researchers is mainly explained by their past publishing experience (supporting the theory of cumulative advantage) or whether some researchers are more productive than others irrespective of their past productivity (supporting the sacred spark hypothesis). Because theories of academic success argue that women have more difficulty accumulating publications than men, we introduce gender as our first variable to explain publications. Concordant with what the literature deems important as mentioned above, we then test whether effects of gender on publications are confounded by access to advanced career stages and international experience, as well as access to better universities. We then test whether effects are confounded by research funding.

Finally, we use fixed-effects regressions (FE) that control for prior productivity, to show what accounts for differences within a career. Fixed effects regressions derive their name from holding all time-invariant differences between individuals constant (fixed) by calculating how each individual diverges from her person-specific mean value, e.g. on publications, awards etc. at each point in time. Thus, FE regressions show how research careers progress within an individual, rather than showing how researchers differ from each other (e.g. how men and women differ), as RE regressions do. As they hold known time-varying and even unknown time-invarying confounders constant, or "fixed" within individuals, FE regressions cannot account for unknown time-varying confounders such as biological age (to the degree that it is not already accounted for by career stages), or for time-varying caring responsibilities (to the degree that these are not accounted for by gender and parental status), or for personal characteristics such as time-varying career orientation. However, using FE regressions does allow us to show who publishes more than his or her prior publication trajectory would lead one to expect, and how this is influenced by access to higher career stages and institutions as well as by research funding.

Controlling for six different types of prior publications, we can control for past achievements to explain current productivity net of past productivity more comprehensively than is typically the case. This addresses the problem that academic performance and motherhood confound each other, as it measures how motherhood impacts performance before, but, crucially, also after controlling for prior performance, thus measuring the effect of motherhood before and after accounting for productivity-based selection into motherhood. Generally, we can account for reciprocal causality in this way, as we a) measure the effect of e.g. mobility and grants on later performance while *not* adjusting for controls, and then measure the same relationship while we do adjust for prior performance. In addition, we use stepwise regressions that follow our theoretical model in introducing first what cannot conceivably come later, e.g. publications can be related to gender, but gender cannot result from publications.

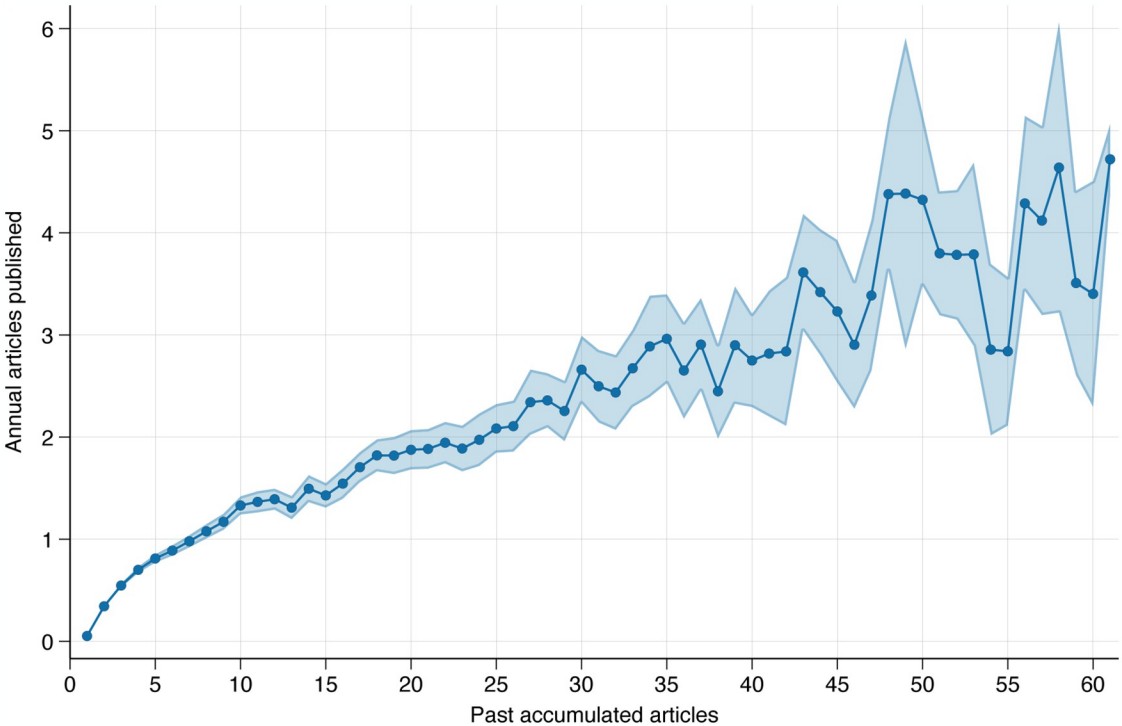

**Fig 1. Relationship between current annual SSCI/SCIE publications and past publications.**

Finally, we use an Oaxaca-Blinder decomposition that shows 1) the extent to which publications are related to gender, 2) whether men publish more because they have more of whatever explains higher productivity or 3) because men publish more because whatever explains higher productivity has a stronger effect on their publications than on those of women. In the following, we begin our analysis descriptively.

## Results: What determines research output?

Fig 1 shows how past and current success are related, by displaying how much researchers publish each year relative to how many SSCI articles they have already published by that year.

Fig 1 shows that researchers with more accumulated publications also publish more in each current year. For example, psychologists who have accumulated seven journal articles in the past, publish one additional paper annually. In contrast, academics who have already published a total of 22 papers in the past, publish approximately two additional articles annually. This suggests that as researchers gain more experience, their annual productivity increases, exactly the pattern one would expect if earlier success breeds later success, as the theory of cumulative advantage suggests.

However, as mentioned in the Introduction, a number of variables may confound this process of cumulative advantage. Fig 2 shows how this is the case for gender, displaying how many peer reviewed journal articles men and women publish annually at successive career stages.

Fig 2 shows how men publish significantly more at each career stage. However, the curves do not move much further away from each other over career stages, suggesting a roughly constant productivity gap between men and women. But does it follow that women increasingly

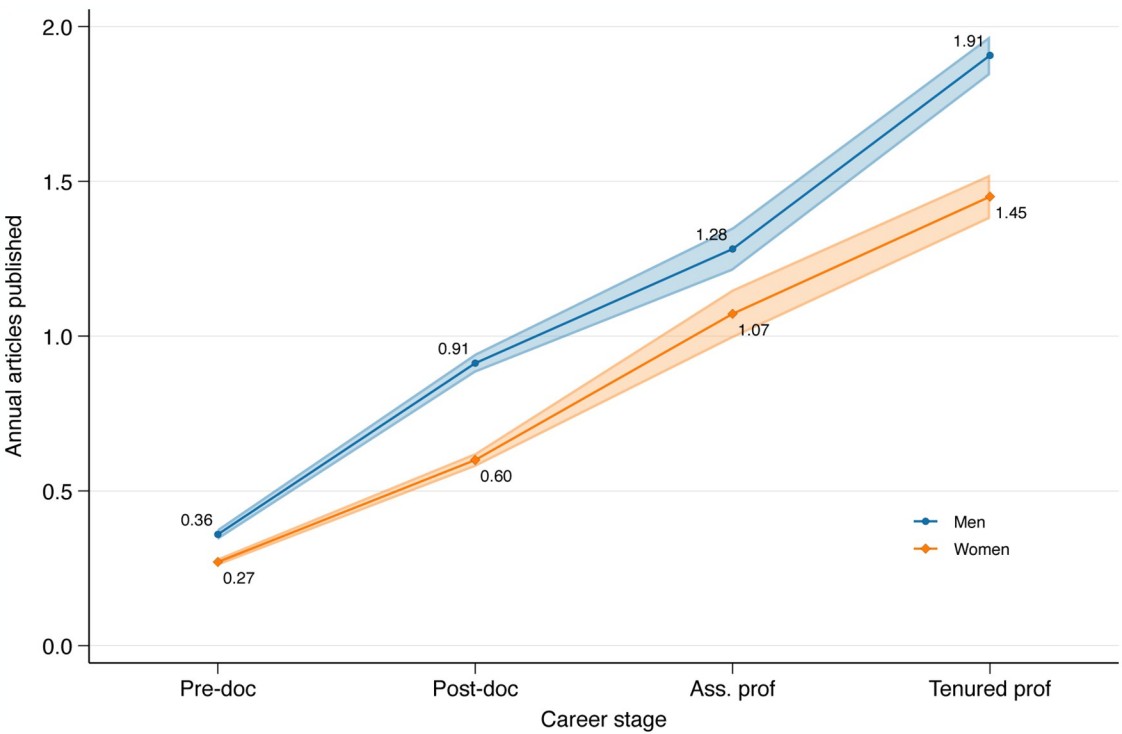

**Fig 2. Annual SSCI/SCIE journal articles, by gender and career status.**

fall behind, as they publish slightly less at each career stage? Fig 3 shows exactly this, displaying the same gender differences in terms of accumulated, rather than annual publications over successive career stages.

Fig 3 shows two curves that do increasingly move apart, suggesting that in terms of accumulated publications, women tend to fall further and further behind over successive career stages.

Taking the evidence of Figs 2 and 3 together, therefore, suggests a process of cumulative advantage: women publish only slightly less at each career stage, yet their productivity increasingly falls behind men over successive career stages, as earlier publications lead to future publications. But how is this confounded by men having access to better institutions or more generous research funding? The following sections use multivariate analyses to disentangle this.

### Random-effects models

The RE models presented below show what explains research productivity, first before (Table 1) and then after (Table 2) controlling for prior productivity. Table 1 shows who publishes more than others, testing the sacred spark hypothesis that some researchers are innately more productive, while the contrast to Table 2 tests the cumulative advantage hypothesis, which posits that those who publish more do so because they have accumulated more publication experience.

Since we are interested in whether the slower accumulation of experience explains why women publish less than men do, we measure differences in numbers of publications both before and after controlling for prior productivity. To understand how this is mediated by access to advanced career stages, institutions and research funding, we successively introduce

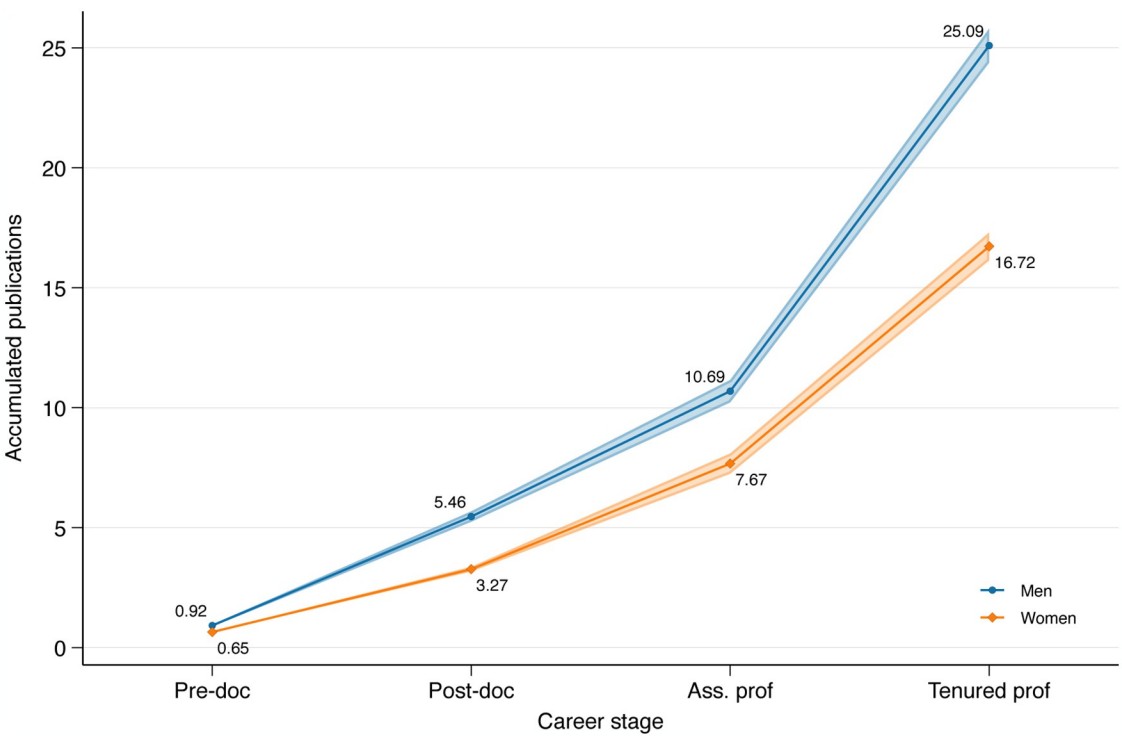

**Fig 3. Cumulative productivity: SSCI/SCIE journal articles by gender and career status.**

these controls, measuring how the female coefficient changes after doing so, which will show how much less women publish if these influences are held constant.

Since we have centered all coefficients on 1, the constant of .58 of Table 1 in Model 1 indicates that during an average year, male researchers with full publication records who belong to the post-2009 cohort publish 58% of what is typical for psychologists in our sample. The "selected publications" coefficient in Model 1 shows that psychologists who only publicize some of their publications have 17% fewer codable publications, which likely means that the dummy variable captures the 17% of publications that these researchers do not show. Importantly, the first variable of Model 1, the female dummy, indicates (before controlling for career steps and other predictors) that female psychologists publish 34% less annually than men do. This is therefore the largely descriptive difference in productivity between men and women that other variables such as accumulated publication experience may explain in the following.

Model 2 adds career stages, measured as pre-doc-, post-doc-, untenured and tenured professor, as well as experience in "universities of excellence" and abroad. Including these factors lowers the productivity gap of women from 34% to 31%. One reason why women publish less than men do is thus that they are less likely to reach advanced career stages, where—as the results show—academics tend to publish more. Notably, post-docs publish 46% more, assistant professors 84% more and tenured professors even 131% more SSCI/SCIE articles annually than psychologists without a PhD. Those who have spent more (log) months outside Germany publish 13% more. Table A6 in the S1 Appendix shows regression models with non-logged coefficients, to illustrate what these results mean in terms of a one-unit increase, indicating that psychologists publish 1% more articles for each month they have spent abroad. Those who have a doctoral degree from abroad do not publish significantly more or less than those who

**Table 1. Random-effects models with annual SSCI/SCIE publications, without controlling for prior publications.**

| | (1) | (2) | (3) | (4) | (5) | (6) |
|---|---|---|---|---|---|---|
| | Baseline | Career | Funding | Parenting | Women only | Men only |
| Female | -0.34*** | -0.31*** | -0.27*** | -0.26*** | | |
| | (-8.80) | (-9.26) | (-8.18) | (-5.17) | | |
| Pre-doc | | 0.00 | 0.00 | 0.00 | 0.00 | 0.00 |
| | | (.) | (.) | (.) | (.) | (.) |
| Post-doc | | 0.46*** | 0.38*** | 0.40*** | 0.35*** | 0.46*** |
| | | (21.87) | (17.59) | (17.84) | (13.97) | (12.65) |
| Assistant prof | | 0.84*** | 0.58*** | 0.61*** | 0.72*** | 0.58*** |
| | | (16.25) | (10.65) | (10.86) | (8.02) | (7.63) |
| Tenured prof | | 1.31*** | 0.72*** | 0.75*** | 0.92*** | 0.70*** |
| | | (19.75) | (9.89) | (9.90) | (9.03) | (6.99) |
| Months abroad (ln) | | 0.13*** | 0.11*** | 0.11*** | 0.08*** | 0.13*** |
| | | (7.93) | (6.99) | (7.02) | (4.76) | (5.22) |
| Doctorate abroad | | -0.07 | 0.03 | 0.03 | 0.12* | -0.04 |
| | | (-1.13) | (0.45) | (0.46) | (1.96) | (-0.43) |
| High-status university | | 0.11** | 0.10** | 0.10** | 0.12** | 0.09 |
| | | (2.90) | (2.78) | (2.89) | (2.81) | (1.37) |
| Research funding (ln) | | | 0.66*** | 0.67*** | 0.47*** | 0.73*** |
| | | | (11.00) | (11.18) | (5.53) | (9.80) |
| Mother | | | | -0.22*** | -0.16*** | 0.00 |
| | | | | (-4.70) | (-3.39) | (.) |
| Father | | | | -0.03 | 0.00 | -0.05 |
| | | | | (-0.45) | (.) | (-0.63) |
| Child = unknown (women) | | | | -0.06 | -0.05 | 0.00 |
| | | | | (-1.58) | (-1.16) | (.) |
| Child = unknown (men) | | | | -0.13* | 0.00 | -0.14* |
| | | | | (-2.17) | (.) | (-2.47) |
| Selected publications | -0.17* | -0.47*** | -0.46*** | -0.46*** | -0.27** | -0.58*** |
| | (-2.28) | (-6.74) | (-6.82) | (-6.97) | (-3.14) | (-6.16) |
| Cohort<1990 | 0.56*** | -0.11 | -0.10 | -0.08 | -0.31*** | 0.02 |
| | (5.94) | (-1.48) | (-1.51) | (-1.18) | (-4.33) | (0.23) |
| 1990–1999 | 0.54*** | -0.01 | -0.07 | -0.06 | -0.09 | -0.02 |
| | (9.36) | (-0.29) | (-1.47) | (-1.21) | (-1.62) | (-0.35) |
| 2000–2009 | 0.39*** | 0.10** | 0.08* | 0.09** | 0.06+ | 0.13* |
| | (10.01) | (3.04) | (2.49) | (2.79) | (1.84) | (2.21) |
| Cohort>2009 | 0.00 | 0.00 | 0.00 | 0.00 | 0.00 | 0.00 |
| | (.) | (.) | (.) | (.) | (.) | (.) |
| Constant | 0.58*** | 0.98*** | 1.00*** | 1.06*** | 0.92*** | 1.15*** |
| | (29.98) | (38.81) | (41.20) | (30.19) | (26.41) | (20.56) |
| R2 within | 0.00 | 0.12 | 0.16 | 0.16 | 0.11 | 0.19 |
| R2 between | 0.14 | 0.35 | 0.39 | 0.40 | 0.39 | 0.37 |
| R2 overall | 0.05 | 0.22 | 0.27 | 0.27 | 0.24 | 0.26 |
| Researcher | 2175 | 2175 | 2175 | 2175 | 1191 | 984 |

(*Continued*)

**Table 1.** (Continued)

|  | (1) | (2) | (3) | (4) | (5) | (6) |
|---|---|---|---|---|---|---|
|  | **Baseline** | **Career** | **Funding** | **Parenting** | **Women only** | **Men only** |
| Observations | 23300 | 23300 | 23300 | 23300 | 10528 | 12772 |

t statistics in parentheses. Variables mean-centered, sd = 1.

[+] p < 0.1,

[*] p < 0.05,

[**] p < 0.01,

[***] p < 0.001.

have a domestic doctoral degree. Psychologists with a degree from a university of excellence publish 11% more than psychologists from other universities.

Model 3 includes research funding, indicating that psychologists with more (log) grants are 66% more productive. Including research grants does not substantially alter the effects of other variables, however, except that it decreases the effect of being a tenured professor. This implies that tenured professors partially publish more because they have acquired more grant money. However, being a woman, having more international experience, or being in a high-status university is unaffected by controlling for research grants.

Model 4 additionally accounts for parental status. It indicates that mothers publish 22% less than childless women, while fathers are not significantly less productive than childless men. Interestingly however, controlling for parental status hardly affects the negative female term, indicating that even though mothers publish significantly less than childless women, this does not explain why childless women publish less. The children non-respondence control is non-significant for women, but significant for men. Finally, Models 5 and 6 calculate separate effects for men and women, suggesting that women profit more from an international doctoral degree and from having been at a "university of excellence."

Table 2 replicates Table 1, but includes past research output in all models. As in Table 1, the coefficients therefore still show differences relative to an average annual research output in psychology, but they now adjust for a psychologist's prior productivity. The models of Table 2 thus show whether variables increase publications because they themselves are related to prior publications or because they make researchers more productive than their prior publication trajectory would lead one to expect.

Model 1 of Table 2 indicates that the number of past (log) SSCI/SCIE publications determines current publications very significantly (t-value of 28) and substantially: academics who published more SSCI/SCIE papers in the past publish 81% more currently, and thus almost twice as much as what is typical for psychology in a given year. The non-logged result of Table A7 in the S1 Appendix shows that for each additional article published in the past, psychologists publish 6% to 8% more articles currently, depending on what is held constant. Those who were highly productive in publishing peer reviewed journal articles in the past are, therefore, also highly productive in the present. However, all other types of prior publication productivity are only insignificantly related to current journal publications. Table 2 Model 1 also shows that female psychologists publish 17% less, even with the same accumulated publication experience. Comparing this to the 34% of Model 1 in Table 1 (which does not account for publication experience), indicates that a process of cumulative disadvantage explains half of the lower female productivity: women publish less than men due to having accumulated less experience in publishing. In contrast, about half of the lower female productivity seems due to women being less productive independently of their past experience.

**Table 2. Random-effects models with annual SSCI/SCIE publications, controlling for prior publications.**

| | (1) | (2) | (3) | (4) | (5) | (6) |
|---|---|---|---|---|---|---|
| | Productivity | Career | Funding | Parenting | Women only | Men only |
| Female | -0.17*** | -0.17*** | -0.17*** | -0.21*** | | |
| | (-6.20) | (-6.27) | (-5.96) | (-4.03) | | |
| Prior SSCI/SCIE articles (ln) | 0.81*** | 0.82*** | 0.75*** | 0.76*** | 0.61*** | 0.75*** |
| | (27.72) | (26.56) | (22.72) | (22.70) | (15.88) | (15.90) |
| Prior monographs (ln) | -0.11+ | -0.08 | -0.08 | -0.08 | -0.00 | -0.14+ |
| | (-1.74) | (-1.28) | (-1.31) | (-1.33) | (-0.00) | (-1.65) |
| Prior book chapters (ln) | -0.05 | -0.03 | -0.03 | -0.03 | -0.00 | -0.05 |
| | (-1.35) | (-0.95) | (-1.01) | (-0.82) | (-0.05) | (-0.99) |
| Prior non-SSCI/SCIE articles (ln) | -0.03 | -0.02 | 0.02 | 0.03 | -0.02 | 0.05 |
| | (-0.69) | (-0.40) | (0.41) | (0.62) | (-0.57) | (0.92) |
| Prior edited volumes (ln) | 0.02 | -0.01 | -0.02 | -0.04 | 0.01 | -0.05 |
| | (0.25) | (-0.12) | (-0.30) | (-0.49) | (0.10) | (-0.50) |
| Prior gray literature (ln) | -0.02 | -0.02 | -0.02 | -0.02 | -0.01 | -0.03 |
| | (-0.65) | (-0.58) | (-0.70) | (-0.58) | (-0.24) | (-0.55) |
| Post-doc | | -0.19*** | -0.18*** | -0.15*** | -0.06+ | -0.09* |
| | | (-6.01) | (-5.86) | (-5.05) | (-1.89) | (-2.36) |
| Assistant prof | | -0.31*** | -0.34*** | -0.31*** | -0.00 | -0.33*** |
| | | (-4.60) | (-5.12) | (-4.66) | (-0.03) | (-3.89) |
| Tenured prof | | -0.23** | -0.37*** | -0.34*** | -0.02 | -0.38*** |
| | | (-3.08) | (-4.63) | (-4.17) | (-0.20) | (-3.80) |
| Pre-doc | | 0.00 | 0.00 | 0.00 | 0.00 | 0.00 |
| | | (.) | (.) | (.) | (.) | (.) |
| Months abroad (ln) | | 0.06*** | 0.06*** | 0.06*** | 0.04* | 0.06*** |
| | | (4.31) | (4.39) | (4.44) | (2.52) | (3.36) |
| Doctorate abroad | | -0.12* | -0.06 | -0.07 | 0.03 | -0.13 |
| | | (-2.02) | (-1.10) | (-1.23) | (0.58) | (-1.54) |
| High-status university | | 0.12** | 0.11** | 0.12** | 0.10* | 0.11* |
| | | (3.15) | (3.16) | (3.26) | (2.55) | (2.19) |
| Research funding (ln) | | | 0.30*** | 0.31*** | 0.18* | 0.36*** |
| | | | (4.62) | (4.81) | (2.16) | (4.42) |
| Mother | | | | -0.27*** | -0.22*** | 0.00 |
| | | | | (-5.74) | (-4.89) | (.) |
| Father | | | | -0.19* | 0.00 | -0.21** |
| | | | | (-2.34) | (.) | (-2.66) |
| Child = unknown (women) | | | | -0.05 | -0.05 | 0.00 |
| | | | | (-1.49) | (-1.38) | (.) |
| Child = unknown (men) | | | | -0.19** | 0.00 | -0.21*** |
| | | | | (-3.13) | (.) | (-3.80) |
| Selected publications | -0.22*** | -0.17** | -0.19*** | -0.19*** | -0.08 | -0.24*** |
| | (-3.96) | (-3.07) | (-3.60) | (-3.60) | (-1.06) | (-3.68) |
| Cohort<1990 | -0.21*** | -0.18** | -0.18** | -0.16** | -0.32*** | -0.05 |
| | (-3.69) | (-3.16) | (-3.23) | (-2.84) | (-4.45) | (-0.76) |
| 1990–1999 | -0.15*** | -0.10** | -0.12** | -0.11** | -0.13* | -0.06 |
| | (-3.65) | (-2.63) | (-3.23) | (-2.90) | (-2.42) | (-1.13) |
| 2000–2009 | -0.02 | 0.02 | 0.02 | 0.03 | 0.03 | 0.05 |
| | (-0.86) | (0.88) | (0.77) | (1.28) | (1.05) | (1.26) |

*(Continued)*

**Table 2.** (Continued)

|  | (1) | (2) | (3) | (4) | (5) | (6) |
|---|---|---|---|---|---|---|
|  | Productivity | Career | Funding | Parenting | Women only | Men only |
| Cohort>2009 | 0.00 | 0.00 | 0.00 | 0.00 | 0.00 | 0.00 |
|  | (.) | (.) | (.) | (.) | (.) | (.) |
| Constant | 1.09*** | 1.05*** | 1.06*** | 1.16*** | 1.04*** | 1.22*** |
|  | (43.20) | (47.56) | (48.65) | (30.30) | (28.43) | (22.87) |
| R2 within | 0.17 | 0.17 | 0.18 | 0.18 | 0.13 | 0.21 |
| R2 between | 0.65 | 0.65 | 0.66 | 0.66 | 0.63 | 0.66 |
| R2 overall | 0.37 | 0.38 | 0.38 | 0.39 | 0.33 | 0.39 |
| Researcher | 2175 | 2175 | 2175 | 2175 | 1191 | 984 |
| Observations | 23300 | 23300 | 23300 | 23300 | 10528 | 12772 |

t statistics in parentheses. Variables mean-centered, sd = 1.

[+] $p < 0.1$,

[*] $p < 0.05$,

[**] $p < 0.01$,

[***] $p < 0.001$.

Model 2 of Table 2 also suggests that post-docs, assistant professors and tenured professors respectively publish 19%, 31% and 23% less than would be expected from an average output trajectory in psychology. Comparing this to the stronger effects of career stages in Table 1 indicates that psychologists publish more at later career stages because they have more experience with publications during these later career stages, rather than because they have reached higher career stages per se. The apparent positive effect of career stages on publications therefore seems to be largely epiphenomenal of the experience in publishing that comes with later career stages. Model 2 of Table 2 also shows that irrespective of their publication experience, those who have spent more months abroad publish more, while those with a foreign PhD publish less than a typical publication trajectory would suggest. Also, those who have their degree from a high-status university publish 12% more than would be expected based on their career stages and prior publication trajectories.

Model 3 adds research funding to the model. Researchers with more funding publish 29% more than otherwise similar researchers. Yet excluding their prior publication record, funded scientists were in any case almost 66% more productive (Model 3 in Table 1); there thus remains a net positive effect of research funding on current publications, but most of the effect of funding on publications can be explained through the publication experience that comes with research funding. The inclusion of grants in the model hardly alters the effects of other variables. Model 4 adds parent status. As this is the full covariate model, Fig 4 visualizes its relevant coefficients.

Holding all other factors constant, women publish 21% less than men, while mothers publish 27% and fathers 19% less than an average publication trajectory would lead one to expect. Research funding and having been at a university of excellence predict current productivity less than prior productivity does, making prior productivity by far the strongest and most significant indicator of current productivity. Model 4 in Table 2 also shows that all variables together explain 66% of productivity differences *between* and 18% of productivity-variation *within* the careers of psychologists.

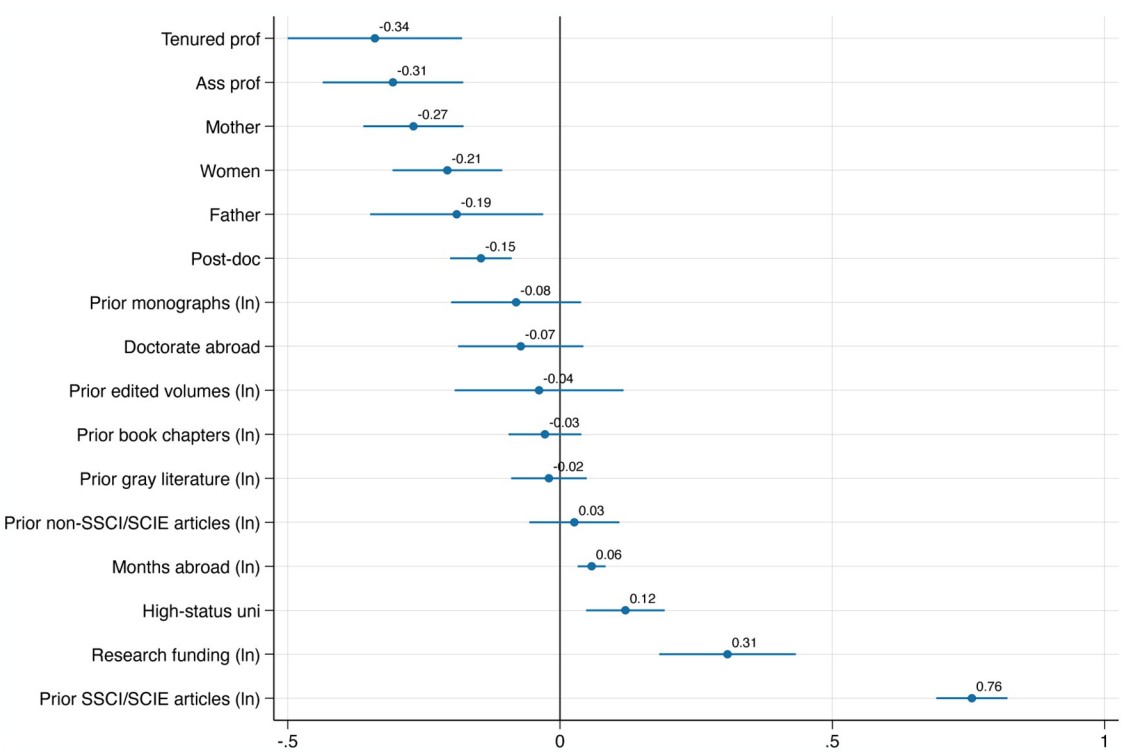

**Fig 4. Visualized effects of Model 4 in Table 2.**

Models 5 and 6 calculate the full Model 4 separately for women and men. Comparing the effect sizes of Model 5 and 6 does not suggest that male and female productivity is influenced by starkly different variables, except that male productivity declines more with career stages.

## Fixed-effects models

Table 3 displays fixed-effects models to explain annual SSCI/SCIE publications, controlling for prior accumulated (log) publications. Coefficients thus indicate under what conditions a psychologist publishes more or less than her or his own typical publication trajectory would lead one to expect, allowing for the most causal interpretation of what influences productivity within a career. Note that this within-career analysis is also a disadvantage of FE models, as these cannot include a female effect or any other time-invariant variable; they only display effects of changes over time within a career, rather than any "between-researcher" effect.

Model 1 of Table 3 indicates that prior SSCI/SCIE articles predict current SSCI/SCIE very significantly. That the effect is lower than in the prior RE Model 1 of Table 2 (.55 vs. .81) suggests that the random effect models partially draw their explanatory power from a between- or population-effect. In other words, the effect of 0.81 partially indicates why some researchers publish *more than others*, while the effect of 0.55 only indicates that *the same psychologist* publishes more than she or he does after having accumulated more publication experience.

Model 2 of Table 3 indicates that psychologists do not publish above their own normal publication rate if (or when) they are a post-doc, assistant professor, or tenured professor. This again implies that reaching higher career stages has no influence on productivity per se. Instead, publications at these career stages merely follow the individual's pre-established productivity trajectory. However, having spent more (log) months abroad increases research

**Table 3. Fixed-effects models with yearly SSCI/SCIE publications, controlling for prior publications.**

| | (1) | (2) | (3) | (4) | (5) | (6) |
|---|---|---|---|---|---|---|
| | Productivity | Career | Funding | Parenting | Women only | Men only |
| Prior SSCI/SCIE articles (ln) | 0.55*** | 0.53*** | 0.39*** | 0.40*** | 0.34*** | 0.43*** |
| | (17.31) | (14.70) | (9.24) | (9.44) | (6.27) | (7.00) |
| Prior monographs (ln) | -0.07 | -0.07 | -0.07 | -0.07 | 0.07 | -0.16 |
| | (-0.91) | (-0.94) | (-0.92) | (-0.95) | (0.86) | (-1.50) |
| Prior book chapters (ln) | 0.06 | 0.06 | 0.05 | 0.06 | 0.12** | 0.02 |
| | (1.48) | (1.53) | (1.28) | (1.51) | (2.63) | (0.41) |
| Prior non-SSCI/SCIE articles (ln) | -0.02 | -0.01 | 0.03 | 0.04 | -0.03 | 0.09 |
| | (-0.38) | (-0.26) | (0.58) | (0.71) | (-0.51) | (1.22) |
| Prior edited volumes (ln) | 0.05 | 0.05 | 0.01 | -0.01 | 0.06 | -0.04 |
| | (0.47) | (0.53) | (0.07) | (-0.09) | (0.51) | (-0.29) |
| Prior gray literature (ln) | -0.02 | -0.03 | -0.03 | -0.03 | -0.09[+] | -0.00 |
| | (-0.50) | (-0.58) | (-0.66) | (-0.75) | (-1.70) | (-0.08) |
| Pre-doc | | 0.00 | 0.00 | 0.00 | 0.00 | 0.00 |
| | | (.) | (.) | (.) | (.) | (.) |
| Post-doc | | 0.01 | 0.07* | 0.09** | 0.07[+] | 0.11* |
| | | (0.32) | (2.10) | (2.84) | (1.80) | (2.37) |
| Assistant prof | | -0.08 | -0.03 | -0.01 | 0.09 | -0.04 |
| | | (-1.06) | (-0.49) | (-0.10) | (0.99) | (-0.48) |
| Tenured prof | | -0.03 | -0.11 | -0.09 | -0.03 | -0.10 |
| | | (-0.47) | (-1.55) | (-1.20) | (-0.32) | (-1.04) |
| Months abroad (ln) | | 0.04* | 0.05* | 0.05* | 0.04 | 0.06* |
| | | (2.13) | (2.46) | (2.51) | (1.35) | (2.04) |
| Doctorate abroad | | 0.00 | 0.00 | 0.00 | 0.00 | 0.00 |
| | | (.) | (.) | (.) | (.) | (.) |
| High-status university | | -0.01 | 0.01 | 0.01 | 0.09 | -0.03 |
| | | (-0.12) | (0.11) | (0.09) | (0.78) | (-0.18) |
| Research funding (ln) | | | 0.44*** | 0.44*** | 0.27*** | 0.49*** |
| | | | (6.84) | (6.83) | (3.49) | (5.82) |
| Mother | | | | -0.33*** | -0.25*** | 0.00 |
| | | | | (-6.03) | (-4.48) | (.) |
| Father | | | | -0.04 | 0.00 | -0.09 |
| | | | | (-0.57) | (.) | (-1.12) |
| Constant | 1.00*** | 1.00*** | 1.00*** | 1.04*** | 0.93*** | 1.11*** |
| | (1.11e+09) | (9.20e+08) | (9.60e+08) | (74.79) | (42.64) | (53.32) |
| R2 within | 0.17 | 0.17 | 0.19 | 0.19 | 0.14 | 0.21 |
| R2 between | 0.62 | 0.62 | 0.59 | 0.59 | 0.52 | 0.58 |
| R2 overall | 0.35 | 0.35 | 0.35 | 0.36 | 0.29 | 0.36 |
| Researcher | 2175 | 2175 | 2175 | 2175 | 1191 | 984 |
| Observations | 23300 | 23300 | 23300 | 23300 | 10528 | 12772 |

t statistics in parentheses. Variables mean-centered, sd = 1.

[+] $p < 0.1$,

* $p < 0.05$,

** $p < 0.01$,

*** $p < 0.001$.

output by 4% above a researcher's prior trajectory. In contrast, spending time at high-status universities does not increase an academic's output above her or his average trajectory. That the effect was positive in the random-effects model suggests that high-status universities have more productive researchers, but going there does not make the same individual more productive.

Model 3 in Table 3 implies that psychologists publish 44% above their individually typical productivity with each log increase in DFG-funded projects. The random-effects models showed a weaker effect, indicating that grants are not simply awarded to more productive academics, but rather that they make the same psychologist more productive than she or he would otherwise have been without funding. This is the opposite of the effect of high-status universities, which attract more productive psychologists, but do not make the same individual more productive.

Model 4 of Table 3 suggests that becoming a mother significantly decreases publication rates, while becoming a father does not. Comparing this to the random-effects models of Table 2 suggests that both mothers and fathers are less productive than childless researchers (holding other influences constant), yet only the productivity of a mother declines below the publication trajectory she had before having children, while a typical father does not publish less after having children.

The last two models estimate the fixed-effects regressions separately for males and females. The results again suggest that when male psychologists have children, they publish only insignificantly less; yet when women have children, they do publish significantly less than they did before. Note also that that the effect of research funding on the publications of men is twice as high as on the publications of women, meaning that men turn the same research funding into more measurable productivity. As is the case with the other statistical relationships, we do not know the causal mechanism behind it.

## Sensitivity tests

**Alternative definition of high-status university.**   We measured the effect of exposure to high-status universities by the proportion of degrees acquired at least once by psychologists from universities that held the title "university of excellence." Different career steps at such universities might, however, have different effects. We therefore disentangled the effect of different career stages, finding that a doctorate from a university of excellence makes the same psychologist about 15% more productive (males 23%, females 4%) while having an assistant professorship or tenure at a university of excellence has no effect. It therefore seems that a doctoral degree from a university of excellence is more beneficial than having passed other career stages there. One reason could be that "universities of excellence" tend to have structured and international doctoral programs ("*Graduiertenkollegs*").

**Oaxaca-Blinder decomposition.**   Our Models 1–4 in Tables 1–3 assume that variables (with the exception of children) have the same influence on men and women. This is why we used Models 5 and 6 in each Table to understand how variables influence men and women differently. This, however, makes it impossible to compare men and women in one model. A three-fold Oaxaca-Blinder decomposition [38, 40] shows a) how much more men publish, but also b) whether this is due to their different attributes, or c) because the same attributes influence men differently than women. A decomposition analysis with our most comprehensive model (Model 4 in Table 2, available upon request) shows that while men publish 1.11 articles annually, women publish 0.67. Of the 0.44 articles that men publish more annually, 0.27 are accounted for by men's different attributes, mainly prior experience in publishing SSCI/SCIE articles. This supports our main conclusion that about half of why women publish less than

men can be attributed to women having less experience in publishing SSCI/SCIE articles early on, which decreases their later productivity, while other effects are minor in comparison.

## Discussion

Prior research shows that human capital, measured through past research output, is a reliable indicator for later output [23; more ambiguously, cf. 4, 41]. This study confirms that past output is a good predictor of later output in German psychology, as psychologists who published more (log) SSCI/SCIE papers in the past also publish about 80% more currently (RE-Model 1 in Table 2). We can therefore explain half of a researcher's current SSCI/SCIE articles from his or her past individual productivity, as measured by SSCI/SCIE articles (FE-Model 1 in Table 3). While at first sight it seems that researchers become more productive with career stages, we could show that productivity is not independently associated with career stages, but rather with the publication experience that comes with these career stages. This suggests that informal "on-the-job training" through publication experience increases future publication productivity, rather than productivity depending on formal career stages such as having a PhD or assistant professorship. This in turn means that the theory of cumulative advantage, which predicts that research output can be explained by a researcher's accumulated publication history, explains productivity better than the sacred spark hypothesis, which argues that some researchers have higher rates of research output, even with similar prior experience. We could also test a number of confounders on this relationship, showing why some researchers accumulate research output at a faster rate.

First, our results indicate a gendered cumulative advantage, in the sense that small differences in research output accumulate to larger differences over time. Notably, our results show that women have about 34% fewer SSCI/SCIE publications overall (Table 1, Model 1), but only 17% fewer articles when controlling for prior publications, experience and research funding (Table 2, Model 3). Therefore, half of the gender gap in research output can be explained through mechanisms of cumulative disadvantage, while the other half is left unexplained. We also find that the same woman publishes 25% fewer SSCI/SCIE-articles once she has a child (FE-model 5 in Table 3), while we find no such effect for fathers.

How does the gendered accumulation of productivity that we find relate to the broader literature? Part of the literature claims that female psychologists catch up with men at later career stages [2]. Our results contradict this. Others have argued that the gender gap in research output may result from a leaky pipeline, e.g. women dropping out of academia [3, 4, 15, 16]. Against this, we do not find that access to career stages is a strongly moderating variable. While controlling for seniority decreases the gender gap from 34% to 31% (Table 1, Models 1–2), controlling for past publication experience decreases the productivity gap to 17%. Also, after accounting for publication experience, adding career stages has no additional effect on reducing the female productivity gap (Models 1 and 2 of Table 2). This means that women may seem less productive because they have not reached higher career stages. Yet what really stands behind the higher productivity that comes with later career stages is the experience in publishing that researchers accrue while getting to those later career stages. Our findings therefore suggest that women do not publish less because they do not reach later career stages, but because they publish less early on. Another prominent explanation for the slower accumulation of success by women is child rearing [4, 10, 18, 19]. Our study confirms this, as women publish fewer SSCI/SCIE articles after having a child, while we find no such effect for fathers.

It has been much discussed in Germany whether singling out "universities of excellence" from other universities makes sense. We find that academics who attend such universities do publish 9% to 12% more SSCI/SCIE articles. However, the fixed effects models show that this

is not because these universities make the same researcher more productive, but because productive academics are more likely to join these universities in the first place [28]. This supports findings from previous studies of US psychology, that elite universities hire scholars who have already built eminence, rather than providing a climate that encourages them to do so [4]. One reason for this may be that "once the basic elements of a true research university [. . .] are in place, additional trappings make relatively little difference" [4]. If this is correct, then our results mean that compared to other German universities, the difference that comes with being a "university of excellence" is not large enough to make researchers more productive. For example, teaching load is legally fixed per career stage in Germany and therefore generally not less of a burden in so-called universities of excellence. Our results suggests that while the German excellence initiative has favored productive institutions, it has not made these institutions more productive.

Our findings also suggest that grants increase publications by 44% above what is expected for a given psychologist (Model 4 of Table 3). This is a relatively strong effect compared to what others have found [32]. In addition, comparing the random and fixed effects models suggests that productive academics do not necessarily get more grants, but that grant-receiving psychologists subsequently *do* publish more than their previous publication trajectory prior to getting a grant would lead one to expect. Giving grants to individuals—in contrast to promoting some institutions to "universities of excellence"–does therefore seem to make researchers more productive, rather than simply rewarding those who were more productive in the first place.

If these effects are taken as guidelines for individual researchers, then they imply that early productivity is a very strong gauge for later potential. Hence if psychologists observe that they publish a lot early on, they can on the whole expect this to continue, as the relationship between early and late productivity is very strong. If, however, one publishes little at the beginning of one's career, our data implies that this low productivity is likely to persist. Researchers are therefore well-advised, when thinking about an academic career, to take their early productivity as a gauge for their overall capability. In terms of advancing female careers, it therefore seems paramount to promote early publications for establishing successful female careers, as early productivity promotes later productivity, leading to a process of cumulative advantage.

At the same time, attending high-status universities or getting a doctorate abroad seems relatively inconsequential for productivity, while spending time abroad or acquiring third-party funding seems to have more influence. A different impression may arise from conflation of between-individual and within-individual effects. For example, German so-called "universities of excellence" do seem to attract more productive psychologists, but—according to our data—do not make researchers more productive once they are there.

Note, however, that these results only apply to psychology and are correlational in nature. While we can show what is related to productivity, we cannot be sure of the causal nature of our relationships. This is because omitted variables could play a role; we cannot, for example, measure career commitment, caring responsibilities or the impact of biological age. Future research therefore has to show whether our results are found in other disciplines and whether a causal effect exists in intervention studies.

## Supporting information

**S1 Appendix.**
(DOCX)

## Author Contributions

**Conceptualization:** Martin Schröder, Mark Lutter.

**Data curation:** Isabel M. Habicht, Mark Lutter.

**Formal analysis:** Martin Schröder, Isabel M. Habicht, Mark Lutter.

**Funding acquisition:** Martin Schröder, Mark Lutter.

**Investigation:** Martin Schröder, Isabel M. Habicht, Mark Lutter.

**Methodology:** Martin Schröder, Isabel M. Habicht, Mark Lutter.

**Project administration:** Martin Schröder, Isabel M. Habicht, Mark Lutter.

**Resources:** Martin Schröder, Mark Lutter.

**Software:** Mark Lutter.

**Supervision:** Martin Schröder, Isabel M. Habicht, Mark Lutter.

**Validation:** Martin Schröder, Isabel M. Habicht, Mark Lutter.

**Visualization:** Martin Schröder, Mark Lutter.

**Writing – original draft:** Martin Schröder.

**Writing – review & editing:** Martin Schröder, Isabel M. Habicht, Mark Lutter.

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
