## [Decision Letter · Decision Letter 0]

19 Jul 2024

PONE-D-24-24540Human capital, research funding, and gender: Determinants of research productivity in German psychologyPLOS ONE

Dear Dr. Schröder,

Thank you for submitting your manuscript to PLOS ONE. After careful consideration, we feel that it has merit but does not fully meet PLOS ONE’s publication criteria as it currently stands. Therefore, we invite you to submit a revised version of the manuscript that addresses the points raised during the review process.

We look forward to receiving your revised manuscript.

Kind regards,

Andrey Lovakov, Ph.D.

Academic Editor

PLOS ONE

Journal Requirements:

Additional Editor Comments:

Below is a summary of the main points raised by the reviewers, together with suggestions for revision. Please address these suggestions and also other suggestions from the reviews below.

- Please clarify whether the study focuses on the sacred spark hypothesis, the cumulative advantage hypothesis, or integrates both. It would be helpful to clearly define the expected empirical patterns and hypotheses for each approach. Additionally, the role of the institutional environment, research funding, and gender should be more clearly addressed.

- Re-evaluate the selection of sources to ensure that they are relevant to the European context. Including more empirical findings from similar national contexts would strengthen the contextual relevance of your study.

- Consider the placement of international mobility within the institutional environment section. Discussing its wider implications, such as network expansion, may provide additional insights.

- Justify the choice of SSCI/SCIE articles and discuss the process of fractionalizing and weighting publications by journal impact factor in the main text.

- Please justify the inclusion of only university psychologists and discuss the potential impact of excluding researchers from other sectors (e.g., MPG institutes).

- A more detailed explanation of your methods and statistical analyses is needed. Including information on how you dealt with reciprocal causality and unknown confounders would be helpful. Ensure that all appendices (D, E, and F) are included in the document.

- Provide more detailed information on how mobility and staying abroad were measured, along with the reliability of these measurements.

- Consider distinguishing between different levels of professorship and career stages to account for heterogeneity within the professor category.

- Think about additional output indicators to capture the impact of research output.

Reviewers' comments:

Reviewer's Responses to Questions

**Comments to the Author**

1. Is the manuscript technically sound, and do the data support the conclusions?

Reviewer #1: Partly

Reviewer #2: Yes

2. Has the statistical analysis been performed appropriately and rigorously? 

Reviewer #1: I Don't Know

Reviewer #2: Yes

3. Have the authors made all data underlying the findings in their manuscript fully available?

Reviewer #1: No

Reviewer #2: No

4. Is the manuscript presented in an intelligible fashion and written in standard English?

Reviewer #1: Yes

Reviewer #2: Yes

5. Review Comments to the Author

Reviewer #1: This is an interesting paper on an important issue, of interest to a large audience (although this is not a formal criterion).

The first problem is that the methods are described rather short, and that it is therefore not completely clear what was done and why. There is no code and no data to check things. A more informative explanation of the various steps in the analysis is required, with information what statistical software was used and with the code. Many things therefore remain somewhat unclear. For example, how was the reciprocal causality (performance <-> motherhood; earlier performance -> mobility (or grants) -> later performance) handled in the analysis. And the random and fixed effect analyses are used among other to control for unknown confounders - what variables not included could still have an effect on the findings, apart from of course disciplinary differences (as the study only includes psychology) and country differences.

Second, the text does refer to appendices A-F, but in the document I downloaded are no appendices D, E, or F.

Third, the information about the data is incomplete. Just as one example, how has mobility and staying abroad been measured? What data were available got that, and how reliable are these? Please describe into more detail how the variables are measured and whether that is reliable.

Some other issues:

- The authors only distinguish three career steps: predoc, postdoc and professor. The last category (and the second too), however, is very heterogeneous and includes everyone from junior/assistant professors to full professors. I would assume that distinguishing junior profs from full profs may have an effect on the outcomes. At least there should have been a convincing argument why this distinction is unimportant.

- The sample includes everyone that got a PhD in psychology since 1980, and that implies a huge variation in age, and even more importantly, in the period they started the career. I would assume that career patterns have changed over time, and that leads to the question how such contextual factors influence the findings. Or the other way around, how is controlled for contextual changes that may have influenced the career patterns?

- Is only the number of publications relevant, or should some other output indicator be preferred, e.g., the number of top cited papers? That would probably be a better indicator, as it includes also the impact of the produced output. And is the analysis sensitive for the selected dependent variable?

Typos (some):

- (p5) It is therefore reasonable to first test whether more [lacking word?] psychologists acquire more funding in the first place ...

- (p11) nale psychologist = male psychologists?

Reviewer #2: The article ‘Human capital, research funding, and gender: Determinants of research productivity in German psychology’ aims to analyze the research productivity of psychologists affiliated with German universities. It focuses on determinants of productivity, including previous success in publishing journal articles, gender, institutional environment, international mobility, and grant funding. The study is based on extensive empirical data, which included coding the CV information of all academic psychologists affiliated with relevant departments in more than 70 German universities. The results of the study confirm the cumulative advantage effect suggesting that previous productivity is the most significant predictor of future productivity. The article demonstrates a thoughtful research design, with longitudinal data providing a substantial advantage for causal inferences, and the statistical analysis has been performed appropriately and rigorously.

However, several comments may help the authors improve the article.

Firstly, the article requires more conceptual clarity. It is unclear whether the focus is on one main determinant or on multiple equivalent factors. Initially, the authors seem to consider two approaches: the sacred spark hypothesis and the cumulative advantage hypothesis. Both approaches conceptualize the prior performance effect differently (as I suppose), though this distinction is not clearly discussed in the theoretical section. While institutional environment, research funding, and gender are important factors that can affect productivity, their roles in the article seem auxiliary as they are considered rather as control variables (“In any case, important intervening variables in any relationship between earlier and later success are a researcher’s institutional environment, research funding and gender, which is why these influences need to be accounted for in any comprehensive model of research productivity”, p.4). This reconstruction is apparent on pages 3-4. However, later in the article, all factors are considered more equivalently, and the initial focus on the clash of two approaches is lost. For example, significant attention is given to gender as a variable. My recommendation is to clarify the role of these factors in the study. It might be beneficial to consider all factors together without focusing on two approaches. If the focus remains on two approaches, it is necessary to clearly define the expected empirical patterns for each approach—specifically, the relationship of the dependent variable with previous productivity or other predictors. Formulating clear hypotheses derived from the two approaches may also bring more conceptual clarity.

It is also unclear how the authors selected sources to cite when discussing the role of various determinants of research productivity given that this topic has been widely researched. Some cited sources are relevant for the American system, which differs significantly from the European context. I suggest that it is important to address the context of the national academic science system, within which the allocation of rewards and recognition of research achievements play significant roles. Authors may bring more empirical findings from similar national contexts.

Furthermore, the placement of international mobility within the institutional environment section may not be appropriate, as the mechanism of mobility may include the expansion of a network of co-authors rather than the effect of institutional place. In other words, such factor as international mobility is broader than institutional environment.

I suggest bring more attention to the discussion of the choice of the main dependent variable. It is worth to put in the main text information from the footnote devoted to the discussion of the two steps: fractionalizing and weighting a publication by journal impact factor (additionally, it seems that the indicated appendix was not placed at the end of the article). There is a consensus in the literature about the hierarchy of publication prestige—papers published in journals of different levels should be regarded as different contribution (see, for example, the discussion in Kwiek, 2023). Given that consensus, it is better to discuss weighting in the main section.

Justifying the choice of SSCI/SCIE articles in the context of selecting the publication type and the Web of Science is also essential. Given the bias towards English-language publications, it is useful to clarify how common it is for German psychologists to publish in international journals. Is it possible to make a career with virtually no such publications? It would help to understand what part of empirical phenomenon authors are studying.

Although time dynamics are accounted for by design, considering the authors' age—both biological and academic—might be worthwhile, as productivity differences may be associated with cohort and specifics of the academic market over different periods.

Although, the article indicates similar or different empirical results in various places, wthen discussing results, it would be beneficial to use a paragraph to place the results in the context of similar studies and discuss any similarities or differences. It is also would be useful to provide any suggestions for revealed differences with previous studies. In this regard, I suggest to pay attention to a recent article with similar findings: "Once highly productive, forever highly productive? Full professors’ research productivity from a longitudinal perspective" (Kwiek, 2023).

6. PLOS authors have the option to publish the peer review history of their article (what does this mean?). If published, this will include your full peer review and any attached files.

Reviewer #1: No

Reviewer #2: **Yes: **Katerina Guba

---

## [Author Response · Author response to Decision Letter 0]

24 Oct 2024

Please the attached R&R document.

---

## [Decision Letter · Decision Letter 1]

11 Dec 2024

PONE-D-24-24540R1Human capital, gender, institutional environment and research funding: Determinants of research productivity in German psychologyPLOS ONE

Dear Dr. Schröder,

I have decided that your manuscript can be accepted for publication. However, I would like to ask you to clarify two questions raised by Reviewer 1 and also to add a short non-technical introduction to the three-fold Oaxaca-Blinder decomposition in the methods section. These are only minor changes and I will not send the manuscript back to the reviewers. I'll look at it myself.

We look forward to receiving your revised manuscript.

Kind regards,

Andrey Lovakov, Ph.D.

Academic Editor

PLOS ONE

Journal Requirements:

Reviewers' comments:

Reviewer's Responses to Questions

**Comments to the Author**

1. If the authors have adequately addressed your comments raised in a previous round of review and you feel that this manuscript is now acceptable for publication, you may indicate that here to bypass the “Comments to the Author” section, enter your conflict of interest statement in the “Confidential to Editor” section, and submit your "Accept" recommendation.

Reviewer #1: (No Response)

Reviewer #2: All comments have been addressed

2. Is the manuscript technically sound, and do the data support the conclusions?

Reviewer #1: Yes

Reviewer #2: (No Response)

3. Has the statistical analysis been performed appropriately and rigorously? 

Reviewer #1: Yes

Reviewer #2: (No Response)

4. Have the authors made all data underlying the findings in their manuscript fully available?

Reviewer #1: Yes

Reviewer #2: (No Response)

5. Is the manuscript presented in an intelligible fashion and written in standard English?

Reviewer #1: Yes

Reviewer #2: (No Response)

6. Review Comments to the Author

Reviewer #1: The comments of the editor and the reviewers have been taken up by the authors, which has improved the paper. However, a few questions remain.

First of all, the text (and earlier papers of the authors) suggest that they deploy an event history analysis. If that is correct it should be said in the methods section, if not the reader would want to have more details about the statistical procedures used.

Secondly, the relation between the fixed effects and the random effects may need some additional clarification. In my understanding (but may e the authors use the FE and RE in a different way, as the terms are used differently in the literature), the difference is that RE method allows for random slopes, Whereas the FE model assumes the same slope for each of the higher order entities. If the RE model has a good fit, can one then still conclude something from the FE analysis?

Thirdly, the three-fold Oaxaca-Blinder decomposition should be introduced in the methods section, as I guess many readers may not know what this is.

I feel that these methodological clarifications are needed, but I think this are only minor revisions for the authors

Reviewer #2: (No Response)

7. PLOS authors have the option to publish the peer review history of their article (what does this mean?). If published, this will include your full peer review and any attached files.

Reviewer #1: No

Reviewer #2: **Yes: **Katerina Guba

---

## [Author Response · Author response to Decision Letter 1]

13 Dec 2024

Response to Reviewers: PONE-D-24-24540: “Human capital, gender, institutional environment and research funding: Determinants of research productivity in German psychology”

We have copied all suggestions below, detailing how we have taken up each. 

Editor comment: I would like to ask you to clarify two questions raised by Reviewer 1 and also to add a short non-technical introduction to the three-fold Oaxaca-Blinder decomposition in the methods section. These are only minor changes and I will not send the manuscript back to the reviewers. I'll look at it myself.

Author response: We have done all of this as mentioned below. 

Reviewer #1: The comments of the editor and the reviewers have been taken up by the authors, which has improved the paper. However, a few questions remain. First of all, the text (and earlier papers of the authors) suggest that they deploy an event history analysis. If that is correct it should be said in the methods section, if not the reader would want to have more details about the statistical procedures used, Secondly, the relation between the fixed effects and the random effects may need some additional clarification. In my understanding (but may e the authors use the FE and RE in a different way, as the terms are used differently in the literature), the difference is that RE method allows for random slopes, Whereas the FE model assumes the same slope for each of the higher order entities. If the RE model has a good fit, can one then still conclude something from the FE analysis? 

Author response: These two points are related, in the sense that the reviewer asks us to explain whether we use event history analysis and to explain which others methods we do use. Event history analysis explains why some individuals are at a higher risk than others of experiencing an event. We have therefore indeed used event history analysis in our papers where we explain who becomes a tenured professor. But we do not use event history analysis here, as we do not explain a (one time) event, but rather the accumulation of publications. Rather than explaining what approach we are not using, we explain the approach that we are using. We have therefore slightly enlarged the section that starts with “Last, we use fixed-effects regressions (FE)…”, essentially explain that RE regressions compare researchers, by drawing from “between-individual” and “within-individual” effects, while FE regressions hold between-individual effects constant (fixed) to only show “within individual” effects.

Reviewer #1: Thirdly, the three-fold Oaxaca-Blinder decomposition should be introduced in the methods section, as I guess many readers may not know what this is. I feel that these methodological clarifications are needed, but I think this are only minor revisions for the author

Author response: Sure, essentially the three-fold Oaxaca-Blinder decomposition shows 1) how much more men publish, 2) whether men publish more because they have more of whatever explains higher productivity or 3) because whatever explains higher productivity has a stronger effect on the publications of men rather than on the publications of women. I have added this explanation to the methods section.

---

## [Editor Report · Decision Letter 2]

19 Dec 2024

PONE-D-24-24540R2Human capital, gender, institutional environment and research funding: Determinants of research productivity in German psychologyPLOS ONE

Dear Dr. Schröder, Thank you for submitting your manuscript to PLOS ONE. After careful consideration, we feel that it has merit but does not fully meet PLOS ONE’s publication criteria as it currently stands. Therefore, we invite you to submit a revised version of the manuscript that addresses the points raised during the review process.

Thank you for resubmitting your manuscript. I am ready to accept your paper. In the cover letter you mentioned that you could send the paper to a professional editor. I am not a native English speaker myself, so I always think that a professional copy editing is a good thing. If you have the resources for it, I would definitely recommend it. I've decided to do a minor revision decision again to give you the chance to go through a professional copy editing. However, no changes (other than copy editing) are required in the text. Once you resubmit the paper, I'll accept it immediately.

We look forward to receiving your revised manuscript.

Kind regards,

Andrey Lovakov, Ph.D.

Academic Editor

PLOS ONE
---

## [Author Response · Author response to Decision Letter 2]

30 Dec 2024

I have nonly changed the main manuscript against a professionally copy edited as required.

---

## [Editor Report · Decision Letter 3]

3 Jan 2025

Human capital, gender, institutional environment and research funding: Determinants of research productivity in German psychology

PONE-D-24-24540R3

Dear Dr. Schröder,

We’re pleased to inform you that your manuscript has been judged scientifically suitable for publication and will be formally accepted for publication once it meets all outstanding technical requirements.

Kind regards,

Andrey Lovakov, Ph.D.

Academic Editor

PLOS ONE
---

## [Editor Report · Acceptance letter]

21 Jan 2025

PONE-D-24-24540R3 

PLOS ONE

Dear Dr. Schröder, 

I'm pleased to inform you that your manuscript has been deemed suitable for publication in PLOS ONE. Congratulations! Your manuscript is now being handed over to our production team.

Kind regards, 

on behalf of

Dr. Andrey Lovakov 

Academic Editor

PLOS ONE